# Once-Weekly Whole-Body Electromyostimulation Increases Strength, Stability and Body Composition in Amateur Golfers. A Randomized Controlled Study

**DOI:** 10.3390/ijerph18115628

**Published:** 2021-05-25

**Authors:** Carina Zink-Rückel, Matthias Kohl, Sebastian Willert, Simon von Stengel, Wolfgang Kemmler

**Affiliations:** 1Institute of Medical Physics, Friedrich-Alexander University of Erlangen-Nürnberg, 91052 Erlangen, Germany; carina.zink-rueckel@imp.uni-erlangen.de (C.Z.-R.); sebastian.willert@imp.uni-erlangen.de (S.W.); Matthias.Kohl@hs-furtwangen.de (S.v.S.); 2Faculty Medical and Life Sciences, University of Furtwangen, 78056 Villingen-Schwenningen, Germany; simon@imp.uni-erlangen.de

**Keywords:** WB-EMS, trunk strength, leg strength, lean body mass, fat mass, low-back pain, hobby golf players

## Abstract

Whole-body electromyostimulation (WB-EMS), an innovative training technology, is considered as a joint-friendly, highly customizable and particularly time-effective option for improving muscle strength and stability, body composition and pain relief. The aim of the present study was to determine the effect of 16 weeks of once-weekly WB-EMS on maximum isometric trunk (MITS), leg extensor strength (MILES), lean body mass (LBM) and body-fat content. A cohort of 54 male amateur golfers, 18 to 70 years old and largely representative for healthy adults, were randomly assigned to a WB-EMS (*n* = 27) or a control group (CG: *n* = 27). Bipolar low-frequency WB-EMS combined with low-intensity movements was conducted once per week for 20 min at the participants’ locations, while the CG maintained their habitual activity. The intention to treat analysis with multiple imputation was applied. After 16 weeks of once-weekly WB-EMS application with an attendance rate close to 100%, we observed significant WB-EMS effects on MITS (*p* < 0.001), MILES (*p* = 0.001), LBM (*p* = 0.034), but not body-fat content (*p* = 0.080) and low-back pain (LBP: *p* ≥ 0.078). In summary, the commercial setting of once-weekly WB-EMS application is effective to enhance stability, maximum strength, body composition and, to a lower extent, LBP in amateur golfers widely representative for a healthy male cohort.

## 1. Introduction

General stability, muscle strength, body composition and pain prevention and/or relief are important goals for mastering everyday tasks and preventing diseases in increased age (e.g., [1]). Resistance exercise favorably affects all of the parameters listed above (e.g., [2,3,4]) and should thus be a relevant component of exercise programs dedicated to health and physical fitness [5]. However, only a minority of people [6,7] achieve the exercise doses recommended for positively impacting muscle mass and function, disabling conditions or obesity [8]. The reasons given for this absence from exercise are predominately time constraints, physical limitations or little enthusiasm for exercise conducted alone [9,10]. Innovative, time-efficient, joint-friendly, supervised and highly individualized exercise technologies should therefore be a good choice for increasing enthusiasm for exercise. Whole-body electromyostimulation (WB-EMS) might be such a candidate. Indeed, recent studies (summarized in [11,12]) have reported favorable results of WB-EMS on muscle strength, body composition and low-back pain, albeit with respect to training frequency as a crucial parameter, not only for the given outcome but also for participants’ acceptance and compliance with the exercise protocol [13], the WB-EMS protocols of scientific studies [11,14] vary considerably from the once-weekly 20 min WB-EMS protocol of commercial WB-EMS facilities. Thus, apart from low-back pain ([12,15], evidence for a positive effect of the commercial “once per week” setting on parameters predominately addressed by the resistance type exercise “WB-EMS” (i.e., stability, muscle strength, body composition) is simply still lacking. Using the vehicle of an ongoing project in amateur/hobby golfers, we aimed to determine the effect of once-weekly 20 min sessions of WB-EMS on muscle strength, body composition and (slightly subordinately) low-back pain in healthy adult males. Of importance for this issue, we selected a cohort of amateur/hobby golfers that widely allows a generalization of the study results to less specific male cohorts.

Our main hypotheses were that 16 weeks of once-weekly WB-EMS application would significantly increase maximum trunk and hip-/leg-extensor strength compared to a corresponding non-training control group.

Our secondary hypothesis was that 16 weeks of once-weekly WB-EMS application would significantly increase lean body mass (LBM) compared to a corresponding non-training control group. Another secondary hypothesis was that 16 weeks of once-weekly WB-EMS application significantly decrease body-fat content compared to a corresponding non-training control group.

Our experimental hypotheses were that 16 weeks of once-weekly WB-EMS application would significantly decrease low-back pain (a) frequency and (b) intensity compared with a corresponding non-training control group.

## 2. Materials and Methods

### 2.1. Experimental Approach

The Franconian EMS and Golf (FrEMGo) study is a randomized controlled exercise trial with a balanced parallel group design and two study arms. The study included male amateur golfers aged 18 to 70 years. Key aims of the trial were (1) to evaluate the effects of 16 weeks of WB-EMS on golf performance and related functional and physical parameters. However, apart from this specific aim, we used this study as a vehicle to assess the effectiveness of one 20 min/week session of WB-EMS on maximum strength and body composition parameters in healthy adult males. However, due to the COVID-19-induced lockdown of golf facilities, we were unable to address the main study outcome “golf performance” as determined by the average result of five 18-hole rounds. Thus, the alignment of this article now widely addresses our second, less specific study aim.

The project was planned and initiated by the Institute of Medical Physics (IMP), University of Erlangen-Nürnberg (FAU), Germany. The University Ethics Committee of the FAU (number 377_19b) approved the trial. The study fully complies with the Helsinki Declaration “Ethical Principles for Medical Research Involving Human Subjects” [16]. After receiving detailed information, all study participants gave their written informed consent. The project was registered under ClinicalTrials.gov: NCT04264416.

### 2.2. Participants

Participants were recruited in December 2019 and January 2020 by personal information, announcements and social media messages. Recruitment was restricted to the mid-western part of Franconia (Bavaria, Germany). Briefly, the inclusion criteria were: (1) men aged 18 to 70 years, (2) with more than two years’ experience in golfing and (3) a golf handicap of 54 or better. Corresponding exclusion criteria were: (1) absolute contraindication for WB-EMS (e.g., arteriosclerosis, electric implants, cardiac pacemakers [17]), (2) contraindications for magnetic resonance imaging, (3) frequent WB-EMS application during the last 12 months, (4) regular resistance exercise for more than 60 min/week during the last 12 months and (5) expected absence of more than one week during the intervention period. In cases of doubt, the study physician decided the eligibility of the participant. Finally, 54 participants were eligible and willing to participate. Participants flow through the study is displayed in Figure 1; baseline characteristics of the participants are listed in Table 1.

Using two strata for age, the 54 participants were randomly assigned to a WB-EMS (*n* = 27) or non-training control (CG: *n* = 27). Participants allocated themselves to the study group by drawing lots from a bowl. The lots were placed in small opaque plastic containers (“kinder egg”, Ferrero, Italy) by a person not involved in the present study. Another person also not involved in the study supervised the drawing procedure. Due to the opaque containers participants were unaware of the result before the drawing procedure. In summary, neither participants nor researchers knew the allocation beforehand (i.e., allocation concealment). After the randomization procedure, the primary investigator (CZ) enrolled participants and instructed them in detail about dos and don´ts. We focused on blinding outcome assessors and test assistants only. Test assessors were not aware of the group status (WB-EMS or CG) of the participants and were not allowed to ask, either.

### 2.3. Interventions

WB-EMS was conducted using a system (miha bodytec^®^, Type II, Gersthofen, Germany) that enables us to simultaneously stimulate thighs and upper arms, hip/bottom, abdomen, chest, lower back, upper back with an overall area of stimulation of about 2600 cm^2^. Of note, the system allows a selectable and thus dedicated intensity for each of the regions. In the present study, we applied a consistently supervised, video-guided WB-EMS program once per week for 16 weeks (mid-January to mid-May). We used bipolar electric current with a frequency of 85Hz, an impulse-width of 350 µs and used an interval approach with 6 s of EMS stimulation with a direct impulse boost and 4 s of rest. Of importance, low-intensity movements or exercises were conducted in a standing position during the 6-s stimulation period [20,21]. While the first 10 min were dedicated to unspecific exercises/movements (see [22]), the last 10 min of the WB-EMS session applied more golf-specific movements (e.g., golf swing, putting movements) (Figure 2.). During the 16-week intervention period, movements/exercise were replaced twice. However, in each case, the intensity of these voluntary movements/exercises per se are not intended to generate muscular effects.

We used a rate of perceived exertion (RPE) to generate a sufficient but tolerable intensity of the EMS application. After four weeks of conditioning with lower impulse intensity, participants were encouraged to exercise at a RPE of 6 to 7 (i.e., hard+ to very hard) on the Borg CR10 Scale [23]. In detail, (impulse) intensity was individually adapted for each body region in close interaction with the participant during the second session and then again after the subsequent sessions were started. However, during the session, instructors slightly increased (impulse) intensity every 3 min in close cooperation with the participants to maintain the prescribed RPE during the session.

We applied a personal training setting with one licensed and experienced instructor responsible for one trainee. Using the mobile medical version of the device, instructors visited participants and conducted the WB-EMS session at the participants’ locations. Due to the COVID-19 pandemic, training was carried out with a strict hygiene concept from week 4 onwards. Individual training appointments were arranged weekly via telephone and WhatsApp communication. Only the trainer and trainees were in the room. In addition, all equipment was cleaned and disinfected before and after each individual WB-EMS application. Each participant received his own WB-EMS garments, which he washed himself. The training took place in compliance with the distance rules and always with a medical mask.

### 2.4. Outcomes

#### 2.4.1. Primary Outcomes

Changes of maximum isometric trunk strength (MITS) from baseline to 16-week follow-up as determined by an isometric test deviceChanges of maximum isokinetic hip/leg extensor strength (MILES) from baseline to 16-week follow-up as determined by an isokinetic leg press

#### 2.4.2. Secondary Outcomes

Changes of lean body mass (LBM) from baseline to 16-week follow-up as determined by a direct segmental multi-frequency bioelectrical impedance analysis (DSM-BIA)Changes of total body-fat content from baseline to 16-week follow-up as determined by a direct segmental multi-frequency bioelectrical impedance analysis (DSM-BIA)

#### 2.4.3. Experimental Outcome

Changes of pain frequency at the lumbar spine from baseline to 16-week follow-up as determined by a pain questionnaireChanges of pain intensity at the lumbar spine from baseline to 16-week follow-up as determined by a pain questionnaire

### 2.5. Changes of Trial Outcomes after Trial Commencement

Due to the temporary closure of golf courses during the COVID-19 pandemic, we are unable to address the intended primary study endpoint, namely, changes in golf performance as determined by an average golf score for five rounds on an 18-hole course.

### 2.6. Assessments

Great emphasis was placed on the standardization of the tests, especially consistent verbal test instructions. All participants were requested to refrain from intense physical activity and exercise 48 h before the assessments. Baseline (January 2020) and 16-week FU assessments (May 2020) were consistently performed by the same research assistant using the same identically calibrated devices, in exactly the same setting and at about the same time of the day (±90 min).

Height was measured using a stadiometer (Holtain, Crymych Dyfed., Great Britain) Body composition was determined by a direct-segmental, multi-frequency bio-impedance analysis machine (DSM-BIA; InBody 770, Seoul, Korea). This device measures impedance of the trunk, arms and legs separately using a tetrapolar eight-point tactile electrode system that applies six frequencies (1, 5, 50, 250, 500 and 1000 kHz). Apart from parameters specific for BIA (i.e., impedance (Z), resistance (R), reactance (XC), phase angle), the device automatically provides total and regional (trunk, extremities) fat and fat-free mass by an equation not published by the manufacturer. In order to standardize the test procedure, participants were requested to refrain from severe physical activity and nutritional intake three hours prior to the DSM-BIA assessment. Reliability of the DSM-BIA device to determine fat-free mass and fat mass was checked by a test–retest protocol in two studies with 2 × 25 male participants 30 to 50 [24] and 70+ years old [25]. Whilst refraining from food, beverages and physical activity, participants were assessed twice within one hour. The resulting intraclass correlation coefficient (ICC) was 0.89 (95% CI: 0.88 to 0.93) and 0.88 (95% CI: 0.85 to 0.91) in the 70+ cohort.

Maximum isokinetic hip/leg extensor strength (MILES) was tested using an isokinetic leg press (CON-TREX LP, Physiomed, Laipersdorf, Germany) at baseline and after 16 weeks. Bilateral hip/leg extension was conducted in a slightly supine (15°) sitting position using hip and chest straps to fix participants. Range of motion of the knee angle was selected between 30° and 90° and the ankle flexed 90° on a flexible sliding footplate. We applied the standard default setting of 0.5 m/s. After a few familiarization movements, we requested participants to conduct five repetitions with maximum effort (“push as strongly as possible”). We conducted two trials with two minutes of rest between the trials. The highest values for hip/leg extension were included in the data analysis. Reliability for the maximum hip/leg extensor strength (test–retest reliability; intra-class correlation) was 0.88 (95% CI: 0.82 to 0.93) for a comparable 30- to 50-years-old male cohort.

Maximum isometric trunk strength was measured as an index of six test exercises: (1) trunk extension, (2) trunk flexion, (3) trunk lateral flexion to the right and (4) trunk lateral flexion to the left side, (5) trunk rotation right and (6) trunk rotation left side. The pelvis and leg area of the test person was fixed in a sitting position. The force transducer was consistently located at the level of the mid-shoulder. Pads lay directly next to the participant without any gaps. Participants were asked to conduct two repetitions intermitted by 30 s of rest with maximum effort (“push as strongly as possible”). The higher value of both trials was included in the data analysis. Results of the tests were summarized and divided by six without any further weighting. Reliability for this test (index) was 0.86 (95% CI: 0.81 to 0.91) for a comparable 30- to 50-years-old male cohort.

Questionnaires and interviews asked for (a) demographic parameters; (b) pain frequency and intensity at the lumbar spine site; (c) diseases, limitations, injuries, and/or operations; (d) pharmacologic agents and/or dietary supplements; and (e) lifestyle including nutritional habits, with special consideration of physical activity and exercise in a dedicated part of the questionnaire [18,26]. The follow-up questionnaires specifically address changes of parameters (i.e., lifestyle, pharmacologic therapy, diseases) that might confound the proper effect of WB-EMS on the study outcome. In order to generate high consistency, completeness and accuracy, the primary investigator (CZ) lastly checked the FU-questionnaire in close interaction with the participants at follow-up.

### 2.7. Statistical Procedures

The sample size of the study was powered on the initially intended primary study outcome “Average golf score of five rounds on an 18-hole course” (see ClinicalTrials.gov: NCT04264416), which, unfortunately, could not be addressed due to the closure of golf courses in the region. However, considering the results of current studies on the parameter addressed in [11,27], the statistical power should be sufficient to address the outcomes of the present study.

All participants initially assigned to the study arms were included in the intention-to-treat (ITT) analysis. R statistics software (R Development Core Team Vienna, Austria) in combination with Amelia II [28] was used to impute missing data in the multiple imputation (ITT). The imputation was repeated 100 times using the full data set for multiple imputations and worked well for all outcomes addressed. The changes over time within groups were analyzed by paired *t*-tests. Time-group interactions (group differences in changes over time, i.e., “effects”) were determined by ANCOVA, adjusting for baseline data using the group as covariate. We applied the approach outlined in Rubin [29] and Barnard and Rubin [30] to calculate the within- and between-imputation variance. In order to adjust for multiple testing (primary outcomes), we applied the Bonferroni–Holm method [31]. Effect sizes were indicated by standardized mean difference (SMD) according to Cohen (d’ [32]). Consistent two-tailed tests were applied and significance was accepted at *p* < 0.05.

## 3. Results

### 3.1. Baseline Characteristics

Table 1 shows baseline characteristics of the WB-EMS and control groups. In summary, randomization and stratification for age was successful with no significant differences between the groups. However, while BMI was comparable between the groups, participants of the WB-EMS were taller and heavier. Furthermore, the proportion of men who conducted resistance-type exercise (only up to 60 min/week) was considerably higher.

### 3.2. Drop out, Attendance and Adverse Effects

Due to the COVID-19 pandemic, we lost several participants to follow-up (Figure 1). Six participants of the WB-EMS quit the study due to the fear of being infected; another participant was considered to have an increased risk of infection. One participant tested positive for COVID-19 during the intervention and decided to quit the study. Finally, one participant of the WB-EMS group was unable to visit the 16-week FU assessment. In parallel, nine participants of the CG were lost to 16-week FU. The majority of the participants (*n* = 7) were unwilling to be assessed due to fear of being infected during the tests; two further participants lost interest and could not be persuaded to attend the tests. Compliance with the WB-EMS protocol as recorded by the instructors was high. Average impulse intensity as prescribed RPE 6 to 7 on a Borg CR-10 scale (i.e., hard+ to very hard) was RPE 6.7 ± 0.5. The average exercise frequency of the participants was close to the prescribed one session per week, due to the possibility to make up a missed session. Apart from periods of muscle soreness, none of the participants reported complaints or adverse effects related to the intervention. Furthermore, no participant reported relevant injuries or diseases during the study period.

### 3.3. Study Outcomes

Table 2 displays the results on main and secondary outcomes. Maximum isometric trunk strength (MITS) increased significantly in the WB-EMS (*p* < 0.001) and decreased slightly in the CG (*p* = 0.142). After 16 weeks of WB-EMS time, group interactions (i.e., effects) were significant (*p* < 0.001) and of high effect size (Table 2).

In parallel, maximum isokinetic hip/leg extensor strength (MILES) increased significantly in the WB-EMS (*p* < 0.001) and increased slightly in the CG (*p* = 0.151). Corresponding between-group differences (“effects”) for MILES were significant (*p* = 0.001); the effect size can be considered as high. Adjusting for multiple testing did not lead to other results (*p* = 0.001). Thus, we confirmed our primary hypothesis that 16 weeks of once-weekly WB-EMS application significantly increase maximum trunk and leg strength in male amateur golfers.

Lean body mass as determined by DSM-BIA decreased significantly (*p* = 0.041) in the CG and increased non-significantly (*p* = 0.21) in the WB-EMS group. Corresponding time–group interactions were significant (*p* = 0.034). Thus, we confirmed our core secondary hypothesis that 16 weeks of once-weekly WB-EMS application significantly affects lean body mass in amateur golfers compared to a non-training control group.

In parallel, body-fat content increased in the CG (*p* = 0.087) and slightly decreased (*p* = 0.382) in the WB-EMS-group. In summary, time-group interaction was not significant (Table 2). Correspondingly, we have to revise our hypothesis on significant WB-EMS effects on body-fat content.

At baseline, only two participants of the CG and three participants of the WB-EMS reported that they did not suffer from back pain at all. The same number of participants (WB-EMS: *n* = 3, CG: *n* = 2) reported frequent (5 on a 7-point scale) to very frequent (6 on a 7-point scale) low-back pain, while none of the participants suffered from chronic low-back pain. In parallel, two participants of the WB-EMS and one participant of the CG reported severe (5 on a 7-point scale) or very severe (6 on a 7-point scale) LBP. During the intervention, LS pain frequency and intensity did not change relevantly in the CG (*p* ≥ 0.825) and fell non-significantly (pain frequency: *p* = 0.078; intensity: *p* = 0.175) in the EG (Table 3). The WB-EMS effect as determined by ANCOVA was not significant (Table 3). Thus, we rejected our experimental hypothesis that 16 weeks of once-weekly WB-EMS application significantly reduce low-back pain (i.e., back pain at the LS) in amateur golfers.

### 3.4. Confounding Parameters

Of surprise during this three-month period of lockdown, no participant of the WB-EMS or control group reported relevant changes of lifestyle including physical activity, exercise or diet. Furthermore, no relevant changes of diseases or general medication were reported. However, the number of subjects with an acute intake of analgesics decreased from five to three participants in the WB-EMS group but was maintained (*n* = 6) in the CG.

## 4. Discussion

The present study is the first RCT to determine the positive effect of the commercial setting of once-weekly WB-EMS on functional and physical outcomes in an adult male cohort. Nevertheless, it should not be neglected that apart from more general fitness and health parameters, the present project initially aimed to address golf-specific outcomes. However, due to the COVID-19-induced inability to determine definite outcome of golf performance, the alignment of the study or rather its interpretation shifted significantly to the second study aim, i.e., the overall effect of 20 mins of WB-EMS once/week on stability, maximum strength, body composition and low-back pain. Hence, although we are unable to decide whether WB-EMS makes amateur golfers better players, the present RCT is nevertheless the first to answer the highly relevant overall question of the effectiveness of the commercial WB-EMS setting on functional and physical outcomes in healthy male adults. It might be surprising, but in contrast to corresponding commercial advertising, the evidence for a positive effect of once-weekly 20-min sessions of WB-EMS is very limited in healthy adults. To the best of our knowledge [11], only three studies address this issue in people with chronic unspecific low-back pain [15,27] or sarcopenic obesity [33]; however, the two studies that focus on this issue show inconsistent results on strength and body composition parameters. In summary, we observed positive effects on MITS and MILES that still remained significant after adjusting for multiple testing of the (two) primary outcomes. In detail, WB-EMS-induced effects on maximum strength changes after 16 weeks (5 to 7%) were moderate at best, but the low training frequency of one session per week and the healthy, moderately fit status of these predominantly young to middle-aged men might put this result into perspective. Indeed, applying the training frequency of 1.5 sessions per week predominately applied in research [11] resulted in higher effects on strength gains (e.g., [11]). Of note, comparable to other WB-EMS studies with a higher WB-EMS application frequency (1.5 to 3 sessions/week; [20,21,33,34,35,36,37]), we observed a significant positive effect on LBM. On the other hand, the effect on body fat content did not reach a significant level in this predominately overweight cohort. However, due to particularities other than the exercise protocol (e.g., age, status, supplementation, assessment), it is difficult to compare the studies in detail. This also applies for the few current WB-EMS trials with similar WB-EMS application (once 20 min WB-EMS/week). Nevertheless, comparable to the present study, the study with sarcopenic obese women [33], which also determined body composition (albeit using dual energy x-ray energy technique), reported significant effects on LBM but low and non-significant effects on total and regional body fat parameters. A corresponding meta-analysis [11] summarizing the WB-EMS effects on body composition in non-athletic adults confirmed the positive effect on muscle mass and the non-significant effect on body-fat parameters; however, there was a substantial level of heterogeneity between the trials results in particular for body fat.

Due to the decision to not apply low-back pain (LBP) as an inclusion criteria, but being fully aware that LBP is a frequent problem in the general adult population [38]), we defined LBP as an experimental study outcome. Indeed, the vast majority of our participants (≈80%) reported suffering from back pain, although the pain frequency and intensity of most participants was low to moderate. The latter finding and the limited statistical power to address this outcome might put the (only) non-significant reductions of pain frequency (*p* = 0.078) and intensity (*p* = 0.152) into a better perspective. However, reviewing the high evidence on WB-EMS effects on low-back pain provided by recent specific studies [15,39,40], this issue might be one of the most clearly confirmed outcomes in WB-EMS research.

At this point, we would like to discuss some study features and limitations, in order to allow the reader to comprehend and interpret our approach. Most relevant and due to the general limitations and the methodologic consequences arising from the lockdown, we only determined moderately strong contributors of golf performance (trunk, leg strength/stability). One might consider the shift from the predominately golf-specific study focus to the also intended, albeit much more general alignment of this contribution as a methodologic flaw. However, from a more pragmatic point of view, it would be not only uneconomical but also morally reprehensible to refrain from a publication of the data with the consequence of the need for a novel study approach. In this context, one may wonder why we did not address further parameters more closely related to golf performance. Indeed, club head speed (CHS), for example, is one such candidate that might be more sensitive [41] to WB-EMS application compared to the much more complex overall result of a round of golf. We aimed to address CHS as the primary study outcome of a twin study in Munich, Germany (clinical trials.gov. NCT04439734) that applied an identical WB-EMS protocol. However, we had to discontinue the intervention after recruitment due to the second COVID 19-induced lockdown in November 2020. Another methodologic limitation might be that our sample size analysis was based on the initial primary outcome average “golf score” that was lost. However, we think it would be inadequate to provide a post hoc analysis for the outcomes addressed here. Although a recognized technique, DSM-BIA is not the gold standard for measuring fat-free/lean body mass [42]. However, due to radiation protection issues, time constraints and the secondary outcome status, we refrained from applying the gold standard for body composition assessment—DXA [42]—to determine body composition in this cohort; there are in fact some concerns about this technology [43,44], particularly in athletes [45]. However, others [46] and we [33,47] observed good to excellent agreements of our device (InBody 770, Seoul, Korea) with DXA (here: Hologic 4500a, Boston, USA) for total lean and fat mass each with narrow limits of agreements. Furthermore, reliability of our DSM-BIA procedure determined for multiple cohorts ranged between (intraclass correlation coefficient: ICC) 0.88 to 0.92 for total body fat, and 0.90 to 0.94 for lean body mass; thus, we think we applied a reliable method for determining body composition. Furthermore, one may feel that drawing lots might not be the most sophisticated randomization approach. However, in the past (e.g., [48,49]), we observed that personally drawing lots and thus randomized self-assignment to a group boosted acceptance for an initially non-favored study group. We included a cohort of healthy 18- to 70-year-old male amateur golf player with a golf handicap below 54. Considering physical activity and exercise, this cohort might be representative for a large amount of the male population; thus, we think our results can be widely generalized. Another particularity of our WB-EMS application was the personal training approach at the participants’ home or location of their choice. Although this setting might not be so far away from the closely supervised one trainer to a maximum of two participants [50] setting respected by most commercial facilities, we feel that adherence to the WB-EMS protocol might be higher than usual. Finally, we are aware that our approach of addressing a complex outcome (“golf performance”) using a less validated intervention (i.e., one session of 20 min/week WB-EMS) is difficult. However, WB-EMS is advertised as a very time saving training technology, and this setting has become established in commercial WB-EMS facilities. Thus, we decided to apply this real-world approach, knowing that this procedure might fail to address golf performance. However, due to the changed alignment of the study, this aspect has become largely irrelevant.

## 5. Conclusions

In summary, we provided evidence for the favorable effects of the commercial WB-EMS setting of one 20-min session/week of WB-EMS on stability, maximum strength, lean body mass and, to a lesser degree, on body fat mass and low-back pain. Although the moment of progression of exercise, or more precisely, impulse intensity is inherently respected by the application of a rate of perceived exertion (e.g., 6 to 7; hard+ to very hard on the Borg CR10 scale [23]) during WB-EMS, compared to other types of exercise, it might be necessary to apply higher training frequencies to avoid plateau effects during sustained application. This should be addressed by longer studies that specifically focus on this issue.

## Figures and Tables

**Figure 1 ijerph-18-05628-f001:**
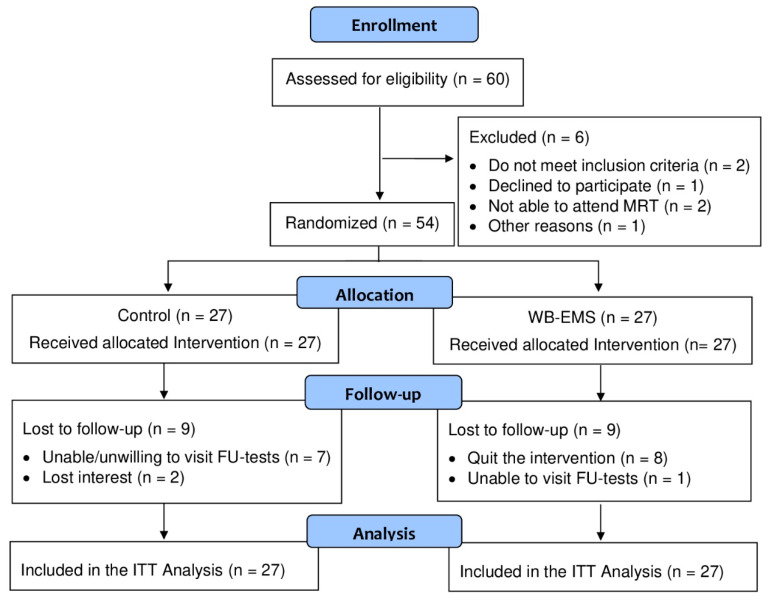
Participant flow through the FrEMGo study.

**Figure 2 ijerph-18-05628-f002:**
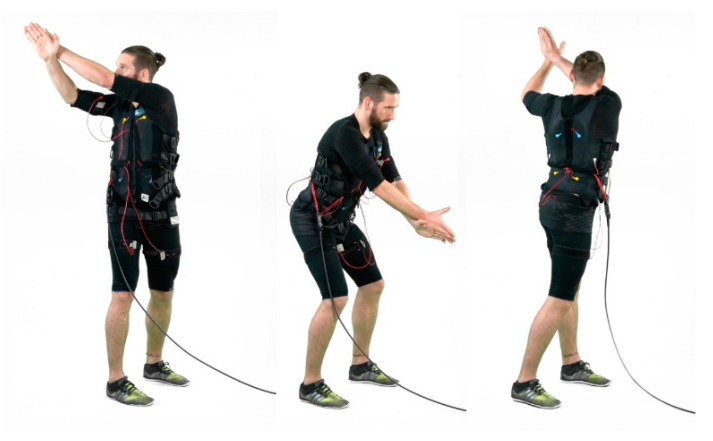
Golf-specific movement during the impulse phase (example). In this context, we focused more on range of motion; thus, we did not instruct participants to consistently watch the ball. Written informed consent was obtained from the participant to publish this picture.

**Table 1 ijerph-18-05628-t001:** Baseline characteristics of the WB-EMS and control group.

Variable	WB-EMS *n* = 27 MV ± SD	Control *n* = 27 MV ± SD	*p*
Age [years]	42.7 ± 16.6	43.0 ± 13.4	0.943
Body Height [cm]	183 ± 8	180 ± 10	0.173
Body Mass [kg]	91.7 ± 17.3	86.1 ± 11.5	0.162
Handicap [Score Points]	16.8 ± 13.7	18.4 ± 14.7	0.694
Physical Activity [Score] ^a^	3.2 ± 1.1	3.5 ± 1.4	0.451
Physical Fitness [Score] ^a^	3.8 ± 1.1	3.8 ± 1.1	0.991
Years Golfing [years]	11 ± 6	10 ± 6	0.529
Frequency golfing [sessions/week]	2.1 ± 1.0	1.8 ± 1.3	0.350
Further exercise [n]	16	12	0.207
Resistance-type exercise [n]	7	2	0.068
Relevant diseases [n]	1	2	0.552
Orthopedic limitations [n]	13	13	1.00
Current smokers [n]	4	4	1.00

^a^ self rated physical activity and fitness; (1: very low to 7: very high) [18,19].

**Table 2 ijerph-18-05628-t002:** Baseline data and changes of primary and secondary outcomes in the WB-EMS and control group.

	CG (*n* = 27) MV ± SD	WB-EMS (*n* = 27) MV ± SD	Difference MV (95% CI)	SMD d’	*p*-Value
Maximum Isometric Trunk Strength Index (MITS)[NM]					
Baseline	182 ± 33	201 ± 38	------------	-------	0.053
Changes	−3.0 ± 9.7	10.7 ± 12.0	13.7 (7.7 to 19.6)	1.26	<0.001
Maximum isokinetic Hip/Leg Extensor Strength (MILES)[N]					
Baseline	3729 ± 889	3581 ± 754	------------	-------	0.528
Changes	57 ± 183	261 ± 245	204 (84 to 324)	0.94	0.001
Lean Body Mass [kg]					
Baseline	66.0 ± 6.7	69.2 ± 10.6	------------	-------	0.192
Changes	−0.54 ± 1.32	0.30 ± 1.33	0.83 (0.07 to 1.60)	0.63	0.034
Body-fat content [%]					
Baseline	23.6 ± 8.5	22.7 ± 6.3	------------	-------	0.679
Changes	0.52 ± 1.50	−0.29 ± 1.73	0.91 (−0.10 to 1.72)	0.50	0.080

**Table 3 ijerph-18-05628-t003:** Baseline data and changes of pain frequency and intensity at the lumbar spine in the WB-EMS and control group.

	CG MV ± SD	WB-EMS MV ± SD	Difference MV (95% CI)	SMD (d’)	*p*-Value
Pain frequency Lumbar Spine [Score-Points] ^a^					
Baseline	2.52 ± 1.55	2.11 ± 1.72	------------	--------	0.365
Changes	0.05 ± 1.03	−0.52 ± 1.55	0.58 (−0.03 to 1.18)	0.43	0.078
Pain Intensity Lumbar Spine [Score-Points] ^a^					
Baseline	2.67 ± 1.51	2.15 ± 1.46	------------	-------	0.207
Changes	−0.01 ± 1.22	−0.39 ± 1.44	0.38 (−0.31 to 1.05)	0.28	0.154

^a^ (1: very low to 7: very high).

## Data Availability

The datasets generated during and/or analyzed during the current study are available from the corresponding author on reasonable request.

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
