# Peer review of "Once-Weekly Whole-Body Electromyostimulation Increases Strength, Stability and Body Composition in Amateur Golfers. A Randomized Controlled Study"

_ijerph, 2021, doi:10.3390/ijerph18115628_

Round 1

Reviewer 1 Report

This study assesses the use of WB-EMS for improving strength, stability and body composition in amateur golfers.

No mention of low back pain is made in the title, however this variable is mentioned in the hypotheses.

The Abstract starts with WB-EMS. This should be written in full as you have done with MITS, MILES etc.

Line 2 of the Introduction add a comma after "only."

Line 37: Change "example for" to "example of" and "might be" to "is."

Line 37-38: Do you have a reference for this statement about players low readiness to invest further resources?

Line 48 & 50: Use "low back pain" or "lower back pain" but not both. Although both are commonly used in the literature, it is best to be consistent throughout your manuscript.

Line 60: Change "application significantly" to "application would significantly."

Line 68-70: You hypothesize an increase in low back pain?

Line 69: Change compared to" to "compared with."

Line 81-82: Change "articles" to "article."

In the first picture of Figure 2 the participant is looking at his hands. If this was designed to replicate a golf swing, surely watching the ball would be appropriate. Why did you instruct participants to look at their hands?

Experimental outcomes include pain. This should be mentioned in the title.

The statistical approach used is appropriate.

The baseline characteristics table should be included in the Methods section rather than the Results section.

Line 241: Delete "as" after "participant" and before "tested."

The results are clearly presented.

If you have included non-golfers in the cohort. You should consider changing the study cohort to active males aged 18-70 years rather than being specific to golfers and then including non-golfers as well.

Again at the beginning of the Discussion section you mention the study outcomes and include low back pain. This seems to be an important measure rather than a secondary one. I think you should mention it in the title and as a primary variable under investigation.

You mention that the low training frequency likely explains moderate changes in strength. Why did you use once per week when exercise interventions of 1.5 times weekly are typically prescribed as you mentioned in line 321?

Line 325-326: Change "once 20 min WB-EMS/week" to "20 mins WB-EMS once/week."

Lines 332-341: Here again you discuss LBP here but you consider ir a secondary aspect of the study. It seems to be quite relevant and I suggest rewriting the manuscript to include LBP as one of the initial main variables of interest.

Reviewer 2 Report

This study investigates the effect of a 16-week EMS intervention on strength, stability and body composition in amateur golfers Although the topic is interesting given the scarcity of data in the literature, there are several aspects that need to be revised.

Title:

-Acronyms should be removed.

-very long.

An alternative could be: The effect of 16 weeks of whole-body electrostimulation on strength, stability and body composition in amateur golfers: a randomized controlled trial

Abstract:

EMS: report the full name at its first mention

Group by time interaction should be reported adjusting for age as a confounding variable.

Keywords:

Remove words already reported in the title

Introduction:

The rationale for the conduct of this study is not very clear. I therefore suggest revising the introduction to make clear why your study is of importance for the field. Furthermore, the rationality of the selected variables is not explained. These should be discussed in order of appearance in the methods and results. Finally in the same order discussed in the discussion section. For example, why is it important to consider body composition in golf? what is the relationships between FM and FFM (or lean body mass) and performance? A clear explanation of what these represent is required (I suggest you consider 2 recent studies: 1)Campa, F.; Toselli, S.; Mazzilli, M.; Gobbo, L.A.; Coratella, G. Assessment of Body Composition in Athletes: A Narrative Review of Available Methods with Special Reference to Quantitative and Qualitative Bioimpedance Analysis. Nutrients 2021, 13, 1620. https://doi.org/10.3390/nu13051620 2) Lukaski H, Raymond-Pope CJ. New Frontiers of Body Composition in Sport. Int J Sports Med. 2021 Feb 23. doi: 10.1055/a-1373-5881. Epub ahead of print. PMID: 33621995.).

The same for the other variables

Methods and results:

First, the authors claim to have conducted a randomized trial. This type of study design should follow the guidelines proposed by the CONSORT statement, which is an evidence-based, minimum set of recommendations for reporting randomized trials.

Among the most important information required in the development of a randomized trial, this manuscript lacks, for example:

  1. the details relating to the inclusion and exclusion criteria of the study participants (a clear understanding of these criteria is one of several elements required to judge to whom the results of a trial apply that is, the trial’s generalizability, applicability and relevance to practice);
  2. the method used to generate the random allocation sequence (authors should provide sufficient information that the reader can assess the method used to generate the random allocation sequence and the likelihood of bias in group assignment);

At this regard, I suggest you check the last updated guidelines proposed by Schulz et al. 2010 (doi: 10.1136/bmj.c332).

For scientific reasons, the sample size for trial needs to be planned carefully. The authors did not perform a power analysis to determine the sample size for the study. Finally, other aspects could be improved, such as the presentation of ANCOVA results in the tables; there are no effect size and probability values relative to the interaction between group and time.

There is a basic need to describe the technical characteristics of 770, InBody, Japan device. What is the calibration method to ensure validity (accuracy and precision) of the bioimpedance measurements? What is the technical error of measurement in vivo? Provide readers with a concise description of what this BIA device measures. In particular, what are the measurements detected by this tool? Do they directly measure the raw bioimpedance parameters (e.g., R, Xc and phase angle)? Again, what equation was used to estimate FM and FFM? Is it an equation developed using the Inbody device or an instrument that works with similar characteristics (frequency and technologies)? If this is not the case, some limitations related to the use of this tool should be discussed considering this recent publication: A Narrative Review of Available Methods with Special Reference to Quantitative and Qualitative Bioimpedance Analysis" Nutrients

Discussion

The discussion section is very descriptive and offers limited comparisons to previous research. Similarly, how do practitioner benefit from that? Again, the discussion section fails to relate the findings to this particular application of interest. Moreover, it is important to consider that body composition parameters are dependent instrument and that the instrumental sensitivities are different. Therefore, no comparisons can be made between studies that estimate body composition with different technologies (e.g., foot-to-hand- or direct segmental in standing position) or sampling frequencies. Authors are therefore encouraged to make substantial changes throughout to improve the overall quality. In the current form the rationale for the study is not clear and I have difficulties finding specific take home messages for practitioners.

Round 2

Reviewer 2 Report

The authors well addressed all my comments and suggestions